# Modeling Clock Comparison Experiments to Test Special Relativity

Xiao-Yu Lu [1], Jin-Shu Huang [1], Cong-Bin Liu [1], Xiu-Mei Xu [1], Jin-Bing Cheng [1], Wan Chang [1], Yu-Yu Zhou [1] and Ya-Jie Wang [2,*]

[1] Henan International Joint Laboratory of MXene Materials Microstructure, College of Physics and Electronic Engineering, Nanyang Normal University, Nanyang 473061, China

[2] Department of Mathematics and Physics, Luoyang Institute of Science and Technology, Luoyang 471023, China

\* Correspondence: yajiewang199373@163.com

**Abstract:** The clock comparison experiments to test special relativity mainly include the Michelson–Morley experiment, Kennedy–Thorndike experiment, Ives–Stilwell experiment and the comparison experiment of atomic clocks in two locations. These experiments can be roughly classified as the comparison of two types of clocks: optical clocks and atomic clocks. Through the comparison of such clocks, Lorentz invariance breaking parameters in the RMS framework can be tested. However, in such experiments, the structural effects of optical clocks have been fully considered, yet the structural effects of atomic clocks have not been carefully studied. Based on this, this paper analyzes the structural effects of atomic clocks in detail and divides the experiments into six types: the comparison of two atomic clocks, two optical clocks, and atomic clocks and optical clocks placed in different and the same locations. Finally, correction parameters for the experimental measurements are given.

**Keywords:** special relativity; Lorentz invariance; clock comparison

**PACS:** 03.30.+p; 04.80.Cc; 06.30.Ft; 07.60.Ly

## 1. Introduction

In 1905, Einstein proposed special relativity (SR) [1], and since then, it has become a major part of modern physics. Lorentz invariance is an important basic assumption [2,3]: the result of any local test experiment is independent of the velocity of the free-falling device, which is considered to be a fundamental symmetry in nature. However, according to the existing grand unified theory [4–6], the Lorentz invariance may be broken, thus, further prompting researchers to test the Lorentz invariance through various experiments [7–9].

There are different theoretical frameworks to study the possibility of Lorentz invariance breaking, here, we discuss only one of them: the kinematics RMS (Robertson–Mansouri–Sexl) framework [10–13], which simply parameterizes the Lorentz transform and limits the deviation between Lorentz invariance breaking parameters and specific theoretical values through experiments. Experiments to test the SR include the Michelson–Morley (MM) experiment [14–16], Kennedy–Thorndike (KT) experiment [15,17–20], Ives–Stilwell (IS) experiment [21–23], and the atomic clock comparison experiments in two different locations [24].

These experiments can well limit the Lorentz invariance breaking parameters in the RMS framework and can finally prove the correctness of SR, where the MM experiment is frequently cited as a direct proof of the light speed invariance. In the MM experiment, a light source emits two beams of light along an orthogonal direction, and after passing through paths of length $L_1$ and $L_2$, respectively, the interference pattern of the two beams of light can be obtained by interferometry.

Then, a new interference pattern can be recorded by rotating the device $90^0$. The upper limit of the Lorentz invariance breaking parameters in the RMS framework can be given by observing the difference between the two interference patterns. We also analyze the MM experiment from another perspective and compare it with other types of experiments. One arm in the interferometer can be regarded as an optical clock, which consists of two mirrors with a distance of $L$ and a photon propagating back and forth between the two mirrors. The period of the light clock is determined by the round-trip time of the light; thus, the flight time of the photon from one mirror to another reflects the frequency of the clock [25–35].

Due to the destruction of the Lorentz invariance in the RMS framework, the light speed in different directions is anisotropic, which affects the frequency of clocks on different detection arms. The MM experiment measures the Lorentz invariance breaking parameters by comparing the clock frequencies in two directions. Analogous to the optical clock, the frequency of the atomic clock is determined by the transition between the two energy levels of the atom.

Since the light speed is anisotropic in the RMS framework and the atom is also a structured particle, the structure of the atom needs to be considered in the process of analyzing the atomic clock comparison. In ref. [36], the structural effects of atoms in the comparison experiments of atomic clocks at different locations are preliminarily considered, but the other classical experiments to test SR are not adequately analyzed, and the complete correction results of each experiment are not given.

In this paper, we classify the MM experiment, KT experiment, IS experiment, and atomic clock comparison experiments in two locations [37–45], which can be regarded as a comparison between two optical clocks composed of cavities and photons, a comparison between an optical clock and atomic clock, and a comparison between two atomic clocks.

Since optical clocks and atomic clocks can be placed in the same location or in different locations, we can roughly divide the experiments into six categories: the comparison of two optical clocks in the same location, the comparison of two optical clocks in different locations, the comparison of optical clocks and atomic clocks in the same location, the comparison of optical clocks and atomic clocks in different locations, the comparison of two atomic clocks in the same location and comparison of two atomic clocks in different locations. Combined with the analysis of atomic structure effects in ref. [36], we comprehensively consider the structure of atomic clocks and optical clocks and analyze the correction of structure effect for the above six types of experiment results.

The paper is organized as follows. In Section 2, we review the typical clock comparison of testing SR under the kinematics RMS framework and expound the current strict restrictions on Lorentz invariance breaking parameters. In Section 3, the frequency shift results are given by analyzing the structural effects of the optical clock in three reference frames for the MM experiment. In Section 4, we analyze the structural effects of the atomic clocks in three reference frames to give the energy levels of the atomic clock, based on which the frequency shift result of the atomic clock is obtained by comparing the differences of clocks for different reference frames. In Section 5, we find the correction of the clock structure effect based on the results of six kinds of experiments. Finally, the paper is concluded in Section 6.

## 2. Clock Comparison to Test Special Relativity Based on RMS Framework

Robertson, Mansouri and Sexl have established a common linear framework by making a simple parameterization of the Lorentz transformation and subsequently experimentally obtaining the deviations of these parameters from the theoretical values in SR. Assuming $\Sigma(T, X, Y, Z)$ is an ideal reference frame, i.e., it is isotropic, and the light speed $c$ is a constant. The cosmic microwave background is usually regarded as an isotropic reference frame, that is, the $\Sigma$ frame.

Then, considering a reference frame $S(t, x, y, z)$ moving at a velocity $v$ relative to an ideal reference frame $\Sigma$, by introducing the Lorentz invariance breaking parameter, the

Lorentz transformation between the laboratory reference frame $S$ and ideal reference frame $\Sigma$ in the RMS framework can be rewritten as [11]

$$
\begin{aligned}
t &= aT + \vec{\varepsilon} \cdot \vec{x}, \\
x &= b(X - vT), \\
y &= dY, \\
z &= dZ,
\end{aligned}
\tag{1}
$$

where $\vec{\varepsilon}$ is determined by clock synchronization, and $v \ll c$ in the laboratory reference frame. Since the one-way light speed is unobservable, all the synchronization conventions are physically equivalent. In this paper, the Einstein synchronization convention $\vec{\varepsilon} = -va(v) / \left[ c^2 \left( 1 - v^2 / c^2 \right) b(v) \right]$ is adopted, and the kinematic parameters $a(v)$, $b(v)$, and $d(v)$ are determined by experiments. Since the $\Sigma$ frame is isotropic, the kinematic parameters $a(v)$, $b(v)$, and $d(v)$ are the functions of $v/c$, which can be written as [46–48]

$$
\begin{aligned}
a(v) &= 1 + \left( \alpha - \frac{1}{2} \right) \frac{v^2}{c^2} + O\left( c^{-4} \right), \\
b(v) &= 1 + \left( \beta + \frac{1}{2} \right) \frac{v^2}{c^2} + O\left( c^{-4} \right), \\
d(v) &= 1 + \delta \frac{v^2}{c^2} + O\left( c^{-4} \right),
\end{aligned}
\tag{2}
$$

where $\alpha$, $\beta$, and $\delta$ represent the Lorentz invariance breaking parameters in the time and space directions, respectively. If $\alpha$, $\beta$, and $\delta$ are all zero, the Lorentz invariance is unbroken. Due to the existence of the Lorentz invariance breaking parameter, there is anisotropy in spacetime. Combining the case of Einstein synchronization with Equations (1) and (2), the expression of the light speed $c(\theta, v)$ in the laboratory reference frame S with respect to the $x$ axis can be obtained as [11]

$$
c(\theta, v) = c \cdot \left[ 1 + (\delta - \beta)\sin^2\theta \frac{v^2}{c^2} + (\beta - \alpha) \frac{v^2}{c^2} \right],
\tag{3}
$$

in which $\theta$ is the angle between the direction of the light propagation and the velocity $v$ of the laboratory reference frame.

At present, the experiments to test the Lorentz invariance breaking parameters in the RMS framework mainly include: the MM experiment, KT experiment, IS experiment, and atomic clock comparison experiments in two locations. The above experiments mainly limit the Lorentz invariance breaking parameters by measuring the light speed in all directions. According to Equation (3), the light speed in two directions for the experimental reference frame is inconsistent, which is related to the combinations of Lorentz invariance breaking parameters $\delta - \beta$ and $\beta - \alpha$. The MM experiment is essentially similar to the Michelson interferometer, since the directions of the two detection arms are coincident, there is an angle $\theta$ modulation.

This experiment mainly limits the combination of Lorentz invariance breaking parameters $P_{\text{MM}} = \delta - \beta$ related to angle $\theta$ by measuring the light speed in both directions, and the strictest limit of this parameter is $|P_{\text{MM}}| \leq (-1.6 \pm 6.0 \pm 1.2) \times 10^{-12}$ [19]. The KT experiment measured another combination of Lorentz invariance breaking parameters $P_{\text{KT}} = \beta - \alpha$ related to the velocity $v$ in Equation (3), and the most stringent limit is currently $|P_{\text{KT}}| \leq (-1.7 \pm 4.0) \times 10^{-8}$ [20]. In addition, the IS experiment limits the Lorentz invariance breaking parameter $\alpha$ in the time direction, and the result is $|\alpha| \leq 2.0 \times 10^{-8}$ [23]. Recently, the comparison of atomic clocks at different locations can also limit the Lorentz invariance breaking parameter $\alpha$, and the result is $|\alpha| \leq 1.1 \times 10^{-8}$ [24].

### 3. Analysis of the Optical Clock Structure Effect with the MM Experiment

The MM experiment was originally an experiment to find ether. A beam of light is split into two beams by a beam splitter, travels a distance of $L_0$ along two mutually orthogonal directions, reflects on two mirrors, and finally recombines at the beam splitter to give the interference fringe. If the light speed is related to the direction, the interference fringe pattern will change with the rotation of the instrument, which can demonstrate the dependence of the light speed on direction. The MM experiment obtains the difference of light speed in different directions through the comparison of the light travel time on two interference arms; from another point of view, the period of light travel reflects the frequency of the light clock, and then each interference arm can be regarded as an optical clock, so the MM experiment is also seen as a frequency comparison between the two optical clocks.

According to the above analysis, the frequency of the optical clock is closely related to the round-trip time of light on the interference arm. Therefore, we can determine whether the speed of light is changed by comparing the round-trip time of light beams on the two interference arms, and then determine whether the Lorentz transformation is broken. Two reference frames are introduced in the RMS framework: the ideal reference frame $\Sigma$ and the laboratory reference frame $S$. In order to facilitate the analysis of the clock frequency in each reference frame, we introduce another ideal reference frame $\tilde{\Sigma}$, which also moves with the velocity $v$ along the $X$-axis relative to the ideal reference frame $\Sigma$. The Lorentz transformation that is satisfied by the two ideal reference frames $\Sigma$ and $\tilde{\Sigma}$ can be described as

$$\tilde{T} = \frac{T - vX/c^2}{\sqrt{1 - v^2/c^2}},$$
$$\tilde{X} = \frac{X - vT}{\sqrt{1 - v^2/c^2}}, \tag{4}$$
$$\tilde{Y} = Y,$$
$$\tilde{Z} = Z.$$

The clock frequency in the laboratory reference frame $S$ can be observed through the connection between the three reference frames, and then the clock frequency correction term arising from Lorentz invariance breaking can be obtained.

For the ideal reference frame $\Sigma$, the light speed remains constant $c$, but the length has a certain contraction, meaning that the travel time of light will change accordingly, which is called the scaling effect. According to Equations (1) and (2), the arm length under the scale effect can be obtained as

$$L = L_0 \left\{ 1 - \left[ \beta - \frac{1}{2} + (\delta - \beta)\sin^2\theta \right] \frac{v^2}{c^2} \right\}. \tag{5}$$

A stationary observer records the propagation time of photons from A to B and then back to A in the ideal reference frame $\Sigma$, which can be written as

$$\Delta T_{AB} = \frac{2L}{c} = [1 - a - \xi(\theta)]\frac{2L_0}{c}. \tag{6}$$

Here, $\xi(\theta)$ reflects the correction related to the structure of the optical clock, which can be described as [36]

$$\xi(\theta) = (\beta - \alpha)\frac{v^2}{c^2} + (\delta - \beta)\sin^2\theta\frac{v^2}{c^2}. \tag{7}$$

According to the above analysis, we obtain that the time recorded by the observer in the ideal reference frame $\Sigma$ has a two-part delay relative to the ideal time $2L_0/c$ from Equation (6): the conventional time delay factor $a$ and the delay arising from the clock structure effect $\xi(\theta)$.

Analogous to the analysis method in the ideal reference frame $\Sigma$, we can calculate the travel time of light in another ideal reference frame $\tilde{\Sigma}$. Although the Lorentz transformation is satisfied between the reference frames $\Sigma$ and $\tilde{\Sigma}$, they are ideal reference frames, so their respective light speeds are $c$ in a vacuum. According to the ref. [36], the arm length of light traveling in the ideal reference frame $\tilde{\Sigma}$ can be obtained as

$$\tilde{L} = L_0 \left\{ 1 - \left[ \beta + (\delta - \beta)\sin^2\theta \right] \frac{v^2}{c^2} \right\}. \tag{8}$$

Further, it can be obtained that the travel time of light in the ideal reference frame is

$$\Delta \tilde{T}_{AB} = \frac{2\tilde{L}}{c} = \left[ 1 - \alpha \frac{v^2}{c^2} - \xi(\theta) \right] \frac{2L_0}{c}, \tag{9}$$

From Equation (9), it can be seen that the time delay obtained by the observer in the ideal reference frame $\tilde{\Sigma}$ contains two items: the delay $1 - \alpha v^2/c^2$ in the time direction and the delay $\xi(\theta)$ caused by the optical clock structure effect. Comparing Equations (9) with (6), they are identical apart from a term, which is proportional to $v^2/2c^2$. The reason for the above results is that the ideal reference system $\tilde{\Sigma}$ and the laboratory reference system $S$ move at velocity $v$ relative to the ideal reference frame $\Sigma$, and there is no relative speed between the reference frame $\tilde{\Sigma}$ and $S$; thus, there is no delay factor $v^2/2c^2$ in the ideal reference system $\tilde{\Sigma}$.

For the laboratory reference frame $S$, the one-way distance traveled by the photon has not changed, i.e., $L_0$. However, the RMS framework assumes that there is a certain break along each direction in this reference frame, meaning that the light speed along each direction is not a constant value, but there is a certain change. The light speed in this reference frame is related to the velocity $v$ and direction $\theta$ of the movement of the laboratory reference frame $S$ relative to the ideal reference system $\Sigma$, which is $c(\theta)$ in Equation (3). Therefore, the travel time of light in the reference system $S$ is

$$\Delta t_{AB} = \frac{2L_0}{c(\theta)} = [1 - \xi(\theta)] \frac{2L_0}{c}. \tag{10}$$

According to the above analysis, it can be seen that there is no relative motion for the observation clock in the laboratory reference frame $S$, so it has only the delay $\xi(\theta)$ arising from the clock structure effect relative to the reference value $2L_0/c$.

As the destruction of Lorentz invariance will cause the speed of light to be anisotropic, and the propagation of light in all directions is used to define the time of the clock, the dependence of experimental results and directions can be tested by comparing the clock frequencies formed in all directions. For the clock comparison experiment, the observer emits the electromagnetic signal at frequency $\nu_1$, and the measured clock frequency is $\nu_2$ after returning after a $2L_0$ distance, so the clock comparison result is

$$\triangle = \frac{\nu_2 - \nu_1}{\nu_0}, \tag{11}$$

where $\nu_0 \equiv c/2L_0$ is the reference frequency, that is, the reference value. The frequency shift result includes four parts, the gravitational redshift effect, the SR effect (the second order of the Doppler effect), and the two items that are contributed by the deviation of SR: the destructive term in the time direction ($\alpha$ related items) and the spatial direction of the destructive pattern ( $\xi$ related items). The relevant items of $\alpha$ have been extensively studied; however, the relevant items of $\xi$ has only been studied in atomic clock comparison experiments at different locations, without combination with the specific analysis of other clock comparison experiments. We specialize in studying frequency comparisons between two optical clocks, two atomic clocks, and two types of clocks at the same and different

positions. Since the MM experiment compares the travel times of two light beams, it can be regarded as a comparison of the frequencies between the two optical clocks.

Thus, it is necessary to convert the travel time of the light in the above calculation results into the form of frequencies. Assuming that the clock frequency recorded by the observer in the ideal reference frame $\Sigma$ is $\nu_\Sigma$, according to Equation (6), it can be obtained as:

$$\frac{\nu_\Sigma}{\nu_0} = \frac{2L_0/c}{\Delta T_{AB}} = 1 + a + \xi(\theta), \tag{12}$$

which is kept to the first order of $v^2/c^2$ based on a Taylor expansion. Similarly, assuming that the clock frequencies recorded by the observers in the ideal reference frame $\tilde{\Sigma}$ and the laboratory reference frame $S$ are, respectively, $\nu_{\tilde{\Sigma}}$ and $\nu_S$, we find:

$$\frac{\nu_{\tilde{\Sigma}}}{\nu_0} = \frac{2L_0/c}{\Delta \tilde{T}_{AB}} = 1 + \varepsilon + \xi(\theta),$$
$$\frac{\nu_S}{\nu_0} = \frac{2L_0/c}{\Delta t_{AB}} = 1 + \xi(\theta), \tag{13}$$

in which $\varepsilon \equiv 1 + \alpha v^2/c^2$ is the breaking parameter in the time direction. Since the parameter $a$ in the RMS framework includes not only the effect of the slowing down of motion clocks in SR but also the breaking of the time direction, the parameter $\varepsilon$ and $a$ of the RMS framework satisfies the relationship $\varepsilon = a/(1 - v^2/c^2)$. According to Equations (12) and (13), the relationship satisfied by clock frequencies of the three reference frames can be concluded as

$$\nu_S = \frac{\nu_{\tilde{\Sigma}}}{\varepsilon} = \frac{\nu_\Sigma}{a}, \tag{14}$$

where higher-order terms are not considered based on the Taylor expansion. In summary, we can obtain the measurement quantities and clock frequency of each reference frame for three observers at rest in their frames, which are listed in Table 1.

**Table 1.** The measurements and corrections in each reference frame.

| Reference Frame | Light Speed | Length | Clock Frequency | Relative Delay |
|---|---|---|---|---|
| $\Sigma$ | $c$ | $L$ | $\nu_\Sigma$ | $a + \xi(\theta)$ |
| $\tilde{\Sigma}$ | $c$ | $\tilde{L}$ | $\nu_{\tilde{\Sigma}}$ | $\varepsilon + \xi(\theta)$ |
| $S$ | $c(\theta)$ | $L_0$ | $\nu_S$ | $\xi(\theta)$ |
| clock frequency relationship | | | $\nu_S = \frac{\nu_{\tilde{\Sigma}}}{\varepsilon} = \frac{\nu_\Sigma}{a}$ | |

## 4. Analyzing the Structural Effects of Atomic Clocks

In Section 3, we analyzed the frequency shift of the clock in each reference frame and concluded that the clock frequency obtained by the observer at each reference frame had a certain delay. The clock frequency observed by the observer in the reference system $\Sigma$ not only includes the time delay factor $\sqrt{1 - v^2/c^2}$ in SR and the time delay $\varepsilon$ (the sum of the two is $a$) due to the Lorentz invariance breaking in the time direction but also includes the system effect $\xi(\theta)$ from the optical clock structure.

For another ideal reference system $\tilde{\Sigma}$, the frequency observed by the observer in this reference system has no time delay of SR. However, the clock frequency in the $\tilde{\Sigma}$ frame includes delays in the time direction $\varepsilon$ and structural effects caused by spatial structure $\xi(\theta)$. For the laboratory reference frame $S$, it considers the speed of light to be constant, so there is only a delay $\xi(\theta)$ caused by structural effects of the optical clock. This result can be obtained from the analysis of optical clock because the photons in the optical clock have a travel distance $L$ as the optical clock is a structured object.

Therefore, when the experiment is performed with an optical clock, the spatial anisotropy originating from Lorentz invariance violation will be introduced through the structure of the optical clock, resulting in the inclusion of $\xi(\theta)$ in the experimental results.

According to the analysis of the optical clock, we can analyze whether the atomic clock has a structural effect.

For the atomic clock comparison experiment, we take the hydrogen-like ion as an example to establish a simple model. The electron moves around the atomic nucleus at high speed, forming a closed spherical shell. Analogous to the MM experiment, the frequency of the optical clock is closely related to the length of the resonator. For the atomic clock comparison experiment, the atomic energy levels of hydrogen-like ions are closely related to the radial distance between the atomic nucleus and the electron.

Therefore, based on the previous work [36], we analyze the atomic energy level change caused by the destruction of Lorentz invariance from the Dirac equation and then theoretically analyze the measurement parameters of the atomic clock comparison experiment. In the MM experiment, the probe arm between two mirrors can be seen as an infinitely deep potential well $V_{\text{Light}}$, so the movement of photons in this potential well can be expressed as the time of the light clock. We can also perform a similar analysis of the atomic structure, the coulomb potential $V_{\text{Atom}}$ of the atom is equivalent to an infinite deep potential well in the MM experiment, and the state transition is equivalent to the time of the optical clock in the MM experiment, where $V_{\text{Light}}$ and Potential $V_{\text{Atom}}$ are

$$V_{\text{Light}} = \begin{cases} 0, 0 \leq x \leq L_0 \\ \infty, x < 0, x > L_0 \end{cases},$$

$$V_{\text{Atom}} = -\frac{Ze^2}{r}. \tag{15}$$

Here, $r$ is the distance from the electron to the nucleus, and $Z$ is the atomic number. In the following, based on the analysis of the atomic clock structure effect in ref. [36], we explore the delay effects of clocks in each reference frame. For the laboratory reference frame $S$, the clock energy level of the atomic clock is

$$E_{Sn} = \left[1 + \left(\alpha - \frac{\beta + 2\delta}{3} - \sqrt{\frac{16\pi}{5}}\frac{\beta - \delta}{3}Y_2^0\right)\frac{v^2}{c^2}\right]\tilde{E}_n^{(0)}, \tag{16}$$

where $Y_2^0 = \sqrt{5/\pi} \cdot (3\cos^2\theta - 1)/4$ is the spherical harmonic function, and $\tilde{E}_n^{(0)}$ is the reference level. According to Equation (16), the frequency shift of the clock can be written as

$$\frac{f_S}{f_0} = \frac{E_{Sn}}{\tilde{E}_n^{(0)}} = 1 + \alpha\frac{v^2}{c^2} - \left(\frac{\beta + 2\delta}{3} + \sqrt{\frac{16\pi}{5}}\frac{\beta - \delta}{3}Y_2^0\right)\frac{v^2}{c^2}. \tag{17}$$

where $f_0$ is the reference frequency related to $\tilde{E}_n^{(0)}$. The second part of the right side of the equation is the delay factor in the time direction, and the third part is the delay effect arising from the structure of the atomic clock. Similarly, the clock frequency shift of the other two reference frames $\tilde{\Sigma}$ and $S$ can be obtained as:

$$\frac{f_\Sigma}{f_0} = \frac{E_{\Sigma n}}{\tilde{E}_n^{(0)}} = 1 + \left(\alpha - \frac{1}{2}\right)\frac{v^2}{c^2} + \alpha\frac{v^2}{c^2}$$

$$- \left(\frac{\beta + 2\delta}{3} + \sqrt{\frac{16\pi}{5}}\frac{\beta - \delta}{3}Y_2^0\right)\frac{v^2}{c^2},$$

$$\frac{f_{\tilde{\Sigma}}}{f_0} = \frac{E_{\tilde{\Sigma} n}}{\tilde{E}_n^{(0)}} = 1 + \left[2\alpha - \left(\frac{\beta + 2\delta}{3} + \sqrt{\frac{16\pi}{5}}\frac{\beta - \delta}{3}Y_2^0\right)\right]\frac{v^2}{c^2}. \tag{18}$$

Combining the second equation in Equation (13) with Equation (17), the frequency shifts of optical clocks and atomic clocks can be obtained as

$$\frac{\nu_S}{\nu_0} = \frac{2L_0/c}{\Delta t_{AB}} = 1 + \left[\beta - \alpha + (\delta - \beta)\sin^2\theta\right]\frac{v^2}{c^2},$$

$$\frac{f_S}{f_0} = \frac{E_{S_n}}{\tilde{E}_n^{(0)}} = 1 + \alpha\frac{v^2}{c^2} - \left(\frac{\beta + 2\delta}{3} + \sqrt{\frac{16\pi}{5}}\frac{\beta - \delta}{3}Y_2^0\right)\frac{v^2}{c^2}. \tag{19}$$

which is the result measured by an observer at rest relative to the laboratory reference system $S$. For the convenience of analysis, utilizing Legendre polynomials to expand spherical harmonics, the above formula can be further simplified as

$$\frac{\nu_s}{\nu_0} = 1 + \left[\frac{1}{3}(\beta + 2\delta) - \alpha + \frac{2}{3}(\beta - \delta)P_2(\cos\theta)\right]\frac{v^2}{c^2},$$

$$\frac{f_s}{f_0} = 1 + \left[\alpha - \left(\frac{\beta + 2\delta}{3} + \frac{2}{3}(\beta - \delta)P_2(\cos\theta)\right)\right]\frac{v^2}{c^2}, \tag{20}$$

with $P_2^0(\cos\theta) = \sqrt{5/\pi}\cdot(3\cos^2\theta - 1)/4 = \sqrt{4\pi/5}Y_2^0$. According to these equations, we can comprehensively analyze the comparison results of the optical clocks and atomic clocks at the same location and different locations and then limit the Lorentz invariance breaking parameters.

## 5. Correction of Clock Comparison Results

In this section, according to the different frequency shifts of optical clocks and atomic clocks, the measurement results of the six kinds of clock comparison can be obtained. Combining Equation (2), we find that $\alpha$ represents the breaking parameter in the time direction, $\beta$ and $\delta$ both represent the breaking parameters in the space direction, where $\beta$ represents the breaking parameter along the motion direction, and $\delta$ represents the breaking parameter in the vertical motion direction.

The MM experiment mainly measures the combination of space parameters $\beta - \delta$, the KT experiment mainly measures the combination of space and time parameters $\beta - \alpha$, and the IS experiment and the comparison experiment of two atomic clocks at the same location mainly measure the time parameter $\alpha$. Based on the above experiments, the comparison between two optical clocks can be divided into two cases: the comparison device is placed in the same location and different locations. For the comparison of two optical clocks at the same location, that is, the MM experiment type, the placement angles $\theta$ of the two optical clocks are inconsistent.

Therefore, according to Equation (20), the test parameter of optical clock comparison at the same location can be obtained as $\beta - \delta$. For the comparison of optical clocks at different locations, their frequencies are $\nu_1$ and $\nu_2$, respectively, so the comparison results will be

$$\frac{\nu_1}{\nu_0} - \frac{\nu_2}{\nu_0} = \left[\frac{1}{3}(\beta + 2\delta) - \alpha\right]\frac{v^2}{c^2}. \tag{21}$$

From the above results, it can be seen that the test parameters are $(\beta + 2\delta)/3 - \alpha$.

Secondly, the comparison of optical clocks and atomic clocks can also be divided into comparisons in the same location and different locations. For the comparison between the optical clock and the atomic clock at the same location, that is, the KT experiment type, according to the analysis of Equation (20), their frequencies of the light clock and atomic clock are $\nu_1$ and $f_1$, respectively. The comparison result can be obtained as

$$\frac{f_1}{f_0} - \frac{\nu_1}{\nu_0} = 2\cdot\left(\alpha - \frac{\beta + 2\delta}{3} + \cdots\right)\frac{v^2}{c^2}, \tag{22}$$

Since $\beta - \delta$ has been limited with a high precision, here, we only consider the constraints of the atomic clock comparison on others. Therefore, the main parameters of the test are $2 \cdot [\alpha - (\beta + 2\delta)/3]$. For the comparison of optical clocks and atomic clocks at different locations, we performed a similar analysis to obtain the main test parameters also as $2 \cdot [\alpha - (\beta + 2\delta)/3]$.

Finally, the comparison between two atomic clocks is also divided into two cases: the device is placed in the same location and different locations. For the comparison between atomic clocks at the same location, their frequencies of the atomic clock are $f_1$ and $f_2$, respectively, that is, the clock comparison result of the IS experiment type is

$$\frac{f_1}{f_0} - \frac{f_2}{f_0} = (\alpha - \frac{\beta + 2\delta}{3})\frac{v^2}{c^2}. \tag{23}$$

Therefore, the main parameters of the test are $\alpha - 2 \cdot (\beta + 2\delta)/3$. Similarly, for the comparison of two atomic clocks in different locations, the main test parameters are the same as the test parameters at the same location.

Comparing the clock comparison results given in this paper based on the analysis of structural effects with the results of several classical experimental tests, the correction of the test results will be obtained. Therefore, the test results have certain corrections as shown in Table 2. Due to the structural effects of the detection arm and atoms, we found that specific corrections were introduced into the original measurement parameters.

**Table 2.** Comparison of the previous results and current revised results of various experiments.

| Experiment Type | Optical Clock and Optical Clock | | Optical Clock and Atomic Clock | | Atomic Clock and Atomic Clock | |
| --- | --- | --- | --- | --- | --- | --- |
| | Same Location (MM Experiment) | Different Location | Same Location (KT Experiment) | Different Location | Same Location (IS Experiment) | Different Location |
| Previous results | $\beta - \delta$ | - | $\beta - \alpha$ | - | $\alpha$ | $\alpha$ |
| Correction results | $\beta - \delta$ | $\frac{1}{3}(\beta + 2\delta) - \alpha$ | $2 \cdot (\alpha - \frac{\beta + 2\delta}{3})$ | $2 \cdot (\alpha - \frac{\beta + 2\delta}{3})$ | $\alpha - \frac{\beta + 2\delta}{3}$ | $\alpha - \frac{\beta + 2\delta}{3}$ |

## 6. Summary

In this paper, we reviewed the main parameters of the SR test by comparing atomic clocks at different locations in the kinematics RMS framework, based on which, the structural effects of optical clocks and atomic clocks were considered to obtain the correction results of the experimental measurement parameters. The Lorentz invariance breaking included not only the breaking in the time direction but also the breaking in the space direction. We found that the theoretic results of the atomic clock experiments included not only the breaking parameters of the time direction but also the breaking parameters of the space direction. Thus, the breaking parameters of the atomic clock comparison experiments in the same place should be corrected to the form of space parameters.

For the analysis of atomic clocks at different locations, due to the existence of structural effects, certain corrections were also introduced to this parameter. In order to better analyze different types of clock comparison experiments and to give the results of clock comparisons at the same location and different locations, we provided the results of different types of clock comparison by calculating the structural effects of optical clocks and combining the structural effects of atomic clocks. Finally, new analysis results were obtained by comparing the parameter combinations in this paper with the original parameter combinations, in which certain correction terms were introduced.

**Author Contributions:** Validation, J.-S.H., C.-B.L., X.-M.X., J.-B.C., W.C. and Y.-Y.Z.; Writing—original draft, X.-Y.L. and Y.-J.W. All authors have read and agreed to the published version of the manuscript.

**Funding:** This work was supported by the Natural Science Foundation of Henan Province (Grant Nos. 232300420354, 23B140013, and 222300420255), and the Nanyang Normal University Science and Engineering Doctoral Program (Grant No. 218294).

**Data Availability Statement:** Not applicable.

**Conflicts of Interest:** The authors declare no conflict of interest.

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
