# Peer review of "Modeling Clock Comparison Experiments to Test Special Relativity"

_universe, doi:10.3390/universe9040189_

Round 1

Reviewer 1 Report

In this manuscript, the possibility of Lorentz-invariance breaking in different types of experimental schemes is investigated. The Lorentz transformation is expressed in the more general RMS framework (Robertson-Mansouri-Sexl), which allows for several “Lorentz invariance breaking parameters” that can be tested experimentally. The authors derive a set of test parameters that can be tested by comparing optical or atomic clock experiments at either the same or different locations.

Overall, the work is interesting and the results are, in my opinion, sufficiently original and significant to be published in Universe. However, the manuscript in the present form lacks clarity and I recommend addressing the following issues before publication:

11.     The authors introduce several terms which are, in some cases, not sufficiently explained/defined or inconsistently used:

-     “ideal reference frame”:  In the second sentence of section II, the ideal reference frame Sigma is introduced, and it is stated that it is isotropic and features constant light speed. I assume, these two properties are the definition of an ideal frame. In this case, this should be unambiguously stated. (This could be solved by just adding an “i.e.”, such as “ … Sigma is an ideal reference frame, i.e., it is isotropic and the light speed c is a constant”). Later (page 6, first sentence) the anisotropic frame S is referred to as an ideal reference frame, too, which appears to be inconsistent with the above definition.

-    “ideal clock”: The concept of the ideal clock is very important for the clock delays derived in section III. There is a short note (above Eq. (11)) saying that the “ideal clock” features a light travel time of 2L_0/c, but no other specifications are given. What is the reference frame of this clock? In a side note (second sentence in section IV) it is stated that a clock in Sigma “has a relative motion with the ideal clock”. I couldn’t find any other information on the reference frame of this clock.

-        “structure”: The term “structure” is explained in some detail for the optical clock referring to an anisotropy arising due to the spatial extension of the “arms” of the clock. There is no explanation of the structure of atomic clocks. Is it due to the polarization of the atomic states, or the direction of field driving the transition, or something else? It would be helpful to give some more details, in particular, to better understand Eq. (15).

-        “test parameter”: In section V, the term “test parameter” is extensively used and it describe the main results of the present work listed in Table II. How are these test parameters defined?

2.      Some of the math given appears to be inconsistent or wrong:

-        Eq. (7) is identical to (6).  I assume the authors want to give an expression for zeta(theta) here, instead.

-        Eq. (11) is claimed to be a frequency, but the expression on the right-handed side corresponds rather to the light travel time (which is the inverse of thee clock frequency).

-        In Eqs. (15) and (16), the position of one of the parentheses is not consistent and should be checked.

-        In Eq. (18) and the following, the symbols v_1 and v_0 are used and not explained. Because the authors refer to Eq. (13), I believe, the actually wanted to use the frequencies nu_0 and nu_1. Is that correct?

3.      Generally, it is not very clear, how the present research advances the field. In Table II, the authors refer to “previous results” without stating where they are coming from (ref {37]?). From my perspective, the significance of the present work hinges on the question, how the results relate to previous results are and why they are (partially) inconsistent. The authors should be as specific and unambiguous as possible addressing this question.

There are several minor issues that I noticed when reading the paper:

-        Abstract: The authors should abstain from using abbreviations without explanations in the abstract (MM, KT, IS, RMS).

-        Section I, second sentence: The authors probably want to say  “More than 100 years after its development…” (The development of SR didn’t take 100 years)

-        The first sentence in the second paragraph does not contain any useful information, and I suggest to remove it.

-        … next sentence: The name of the Physicist is Sexl (not “Sexi”)

-        Page 2, bottom line: “anisotropy in all direction”. I suggest removing “in all directions”.

-        Page 4, in the paragraph after Eq. (9): “Combined with Eq. (2) …”. This sentence is very confusing. I believe, the authors want to say that Eqs.(6) and (9) are identical apart of a term, which is proportional to v^2/2c^2.

-        Summary: The authors summarize “Finally, new measurement results are obtained …”. This statement is very misleading, because the authors did NOT do any measurements.

Author Response

We are very grateful for the reviewers' comments and made necessary revisions point by point based on these constructive suggestions. Please see the attachment for details.

Reviewer 2 Report

Dear Authors: I thing your article is interesting. However, I strongly suggest to put some numbers. What can we experimentally expect form the new results in table II, for example? Also, Sexi should be Sexl (I do  not know Prof. Sexi). There are some sentences I was anable to unerstand. Please, see my report.

Author Response

(The authors gave the same response as above.)

Round 2

Reviewer 1 Report

I have read the revised manuscript and the authors addressed most of the referees’ suggestions and corrections adequately. Therefore, I recommend the publication of the manuscript. However, I still have some minor comments, which the authors might want to consider before publication.

Section I, second sentence: I am quite puzzled, why the authors insist that the development of special relativity took a “long period”. A. Einstein was only twenty-six years old when he published his famous paper in 1905. Some historians estimate that he was working on SR for about seven years. In my opinion, this is a surprisingly short development period for such a ground-breaking and revolutionary theory. I reckon, the authors want to emphasize related developments (for instance the RMS formalism) that took place after the actual development of SR. I would agree that the scientific community had quite a long time to “process” special relativity. But the statement that the development of SR took a “long period” seems very odd to me.

In the first paragraph of section IV, I recommend stating specifically the frame that the “ideal clock” is attached to (although this is in principle clear from the discussion in section III).

The authors should be very careful not to mix up the Roman letter v (typically for speed or velocity) and the Greek letter \nu (for frequency). In the first line of page 5 (just below Table I), this is an expression for \nu_0 (instead of v_0, as stated).

I might have overlooked it, but I didn’t see a definition of f_0 (which is, I believe, the frequency of the “ideal” atomic clock). 

I still miss a definition “test parameter”, which is not identical to the explanation of the parameter alpha, beta, and delta (which easily can be seen form Eq. 2). My understanding is that the given test parameters simply show, what comparison of experiments is sensitive to which of the Lorentz breaking parameters (or algebraic expressions containing several of the breaking parameters). This should be stated in some form in the first paragraph of section V.

Author Response

Thank you for your constructive suggestions for our manuscript. We revised our article based on your comments and please see the attachment for details.
